# Validity and Reliability of the Staden Schizophrenia Anxiety Rating Scale

**DOI:** 10.3390/diagnostics12040831

**Published:** 2022-03-25

**Authors:** Werdie Van Staden, Antonia Dlagnekova, Kalai Naidu

**Affiliations:** 1Centre for Ethics and Philosophy of Health Sciences, Faculty of Health Sciences, University of Pretoria, Pretoria 0002, South Africa; antonia.dlagnekova@up.ac.za; 2Department of Psychiatry, Faculty of Health Sciences, University of Pretoria, Pretoria 0002, South Africa; kalaivani.naidu@up.ac.za

**Keywords:** schizophrenia, anxiety, psychosis, assessment, psychometry

## Abstract

In schizophrenia, none of the standard anxiety measures exhibit strong psychometric properties, and all performed poorly against quality assessment criteria. Developed for the schizophrenia population, this study examined the validity and reliability of the Staden Schizophrenia Anxiety Rating Scale (S-SARS) that measures both specified and undifferentiated anxiety. Among 353 schizophrenia patients, strong correlations with anxiety parameters supported the S-SARS’s convergent validity. Criterion-related validity testing yielded accuracy, sensitivity, and specificity rates of around 95%. Its discriminant validity was observed for measures of depression, psychosis, akathisia, fatigue, vigour, procrastination, behavioural inhibition and activation, and personal growth and initiative. Structural validity was found in a single-factor unidimensional model with a 0.953 factor score. Excellent results were found for internal consistency (Cronbach’s alpha = 0.931; Spearman–Brown coefficient = 0.937; Guttman split-half coefficient = 0.928) and inter-rater reliability (Krippendorff’s alpha = 0.852). It incurred no more than a small error of measurement whereby the observed scores were within 1.54 to 3.58 of a true score on a zero to 50 scale. These strong psychometric properties suggest that the S-SARS is a valid and reliable instrument for measuring specified and undifferentiated anxiety in schizophrenia, providing the means for the accurate measurement of anxiolytic treatment effects.

## 1. Introduction

Anxiety in schizophrenia is common and may present concurrently both as a specified anxiety disorder and in an undifferentiated way [1]. The pooled prevalence rate of specified anxiety disorders in schizophrenia was 38.3% in a meta-analysis of 52 studies, which was higher than the 28.8% reported for the general population [2,3]. In another study, the prevalence was 45% in schizophrenia patients compared to 16% among controls [4]. Prevalence rates for the individual specified anxiety disorders in schizophrenia were previously reported as 14.9% for social phobia, 9.8% for panic disorder, 10.9% for generalised anxiety disorder, 12.1% for obsessive-compulsive disorder, and 12.4% for post-traumatic stress disorder.

Undifferentiated anxiety may be understood as a constellation of clinically significant anxiety features that do not meet diagnostic criteria for any of the specified anxiety syndromes, for which the DSM-IV and DSM-5 respectively provide an anxiety disorder not otherwise specified and an unspecified anxiety disorder [5,6]. Undifferentiated anxiety may also be understood as an anxiety feature that is not specific to any one specified anxiety disorder [1]. Palpitations, hypervigilance, restlessness, trembling, and feelings of tension are examples. In contrast, panic attacks, obsessions, compulsions, specific fears, and excessive worries are more specific to the specified anxiety disorders. Undifferentiated anxiety may feature in the presence or absence of the specified anxiety disorders. For example, a patient may have clinically significant palpitations of anxiety yet not suffer from panic disorder, generalised anxiety disorder, or any of the other specified anxiety disorders. The prevalence rate of undifferentiated anxiety in schizophrenia is not known, but a preliminary suggestion was that 36% of schizophrenia patients had undifferentiated anxiety in the absence of any of the specified anxiety disorders [1].

Measuring instruments of anxiety have been found wanting in the schizophrenia population [7]. A recent systematic review of anxiety measures in non-affective psychoses, not specifically schizophrenia, identified 11 studies evaluating the psychometric properties of 17 instruments [8]. All the evaluated instruments performed poorly against standardised quality assessment criteria, and no single measure of anxiety was considered to demonstrate strong psychometric properties and adequate methodological quality for people with psychosis.

The review nonetheless identified nine anxiety scales that showed acceptable (albeit not strong) psychometric properties. Four of these are the DSM-based Generalised Anxiety Disorder Symptoms Severity Scale (DGSS), Liebowitz Social Anxiety Scale (LSAS), Obsessive–Compulsive Inventory (OCI), and Yale-Brown Obsessive Compulsive Scale (Y-BOCCS), which all focus on specified anxiety (i.e., generalised anxiety, social anxiety, and obsessive-compulsive anxiety). The conceptual scope of the other measures is also limited. The Perseverative Thinking Questionnaire (PTQ) focuses on repetitive negative thinking, the Beck Anxiety Index (BAI) on physiological and cognitive symptoms of anxiety, the Psychological Stress Index (PSI) on stressful life events, and the anxiety subscale of the self-reported Depression Anxiety Stress Scale (DASS) on physiological hyperstimulation and subjective consciousness of anxious affect.

The conceptual scope of the Scale of Anxiety Evaluation in Schizophrenia (SAES) is broader and includes items that are arguably expressions of anxiety, including derealisation, indecision, and pain, but it excludes compulsions. Although the SAES was designed for the schizophrenia population, all its items were taken from existing anxiety scales, and, accordingly, it does not account specifically for the anxiety that is expressed within delusional content and in disturbances of perceptions. The importance of accounting for these central features of schizophrenia is, for example, underscored in the clinical observation that two patients may experience similar threats to their lives as part of persecutory delusions, but one may be intensely fearful and worried and the other not. The same applies for perceptual disturbance; one patient may be fearful and worried about experiencing threatening verbal hallucinations, whereas another who is experiencing these similarly is not.

The underwhelming psychometric properties of the standard anxiety scales in the schizophrenia population may be an expression of clinical complexity in this population. Anxiety may be rather difficult to assess during an acute phase of schizophrenia owing to the psychotic symptoms of this phase [9,10]. Anxiety may further be clinically difficult to distinguish from akathisia, which is a common extrapyramidal side effect of antipsychotic medication [11,12]. Compounding this complexity further, psychotic features and akathisia may exacerbate anxiety and vice versa [9,10,11,12]. Concurrency of comorbid depressive features also complicates assessments as depressive features correlate with anxiety both in the general [13] and schizophrenia populations [14,15,16].

To address the shortcomings of the abovementioned measures, the following objectives for measuring anxiety validly and reliably in schizophrenia may be inferred: It should account for the anxiety that is expressed within delusional content and in disturbances of perceptions, yet anxiety should be discerned from these features of schizophrenia. Moreover, it should account for both the various specified kinds of anxiety as well as undifferentiated anxiety. For measuring anxiety in schizophrenia with these objectives, the Staden Schizophrenia Anxiety Scale (S-SARS) was developed as an assessment and measurement instrument for clinicians to administer during a clinical interview guided by key questions. The S-SARS was used in three previously published studies in which reporting on its psychometric properties was not the aim [1,9,17].

In contrast, this article analyses the data from these studies, separately and pooled, in reporting on the validity and reliability of the S-SARS in both acute and residual phases of schizophrenia.

## 2. Methods

### 2.1. Participants

All participants were 18 years or older and recruited to research participation at a large public sector psychiatric hospital in South Africa contributing to three data sets between the years 2010 and 2019 [1,9,17]. Participants for the first two data sets were inpatients within 10 days of being hospitalized. They met DSM-IV diagnostic criteria for schizophrenia in the acute phase [5]. The acute phase was further defined by the requirement of 60 or more score on the Structured Clinical Interview for the Positive and Negative Syndrome Scale (SCI-PANSS) [18]. Additionally, a score of four (i.e., moderate) or more was required on any two of the SCI-PANSS items that constitute psychotic items, namely, those that measure delusions, hallucinatory behaviour, conceptual disorganisation, or suspiciousness; participants had to be capable of giving informed consent to participate in research, which was captured and affirmed in an informed consent document.

In averting potential confounding influences on anxiety, the following exclusion criteria were applied to the first two data sets: No patients were included who had taken a benzodiazepine during the 24 h preceding the application of the measuring instruments, or for whom zuclopenthixol acetate was administered less than 72 h prior to applying the measuring instruments. Further exclusion criteria were patients with neurological conditions affecting the central nervous system, a known other medical condition that influenced their mental state, or substance dependence (excluding nicotine) by DSM-IV criteria.

The third data set comprised of outpatients attended at the same hospital with residual avolitional schizophrenia [17]. For inclusion, participants had to be diagnosed with schizophrenia in partial remission as defined by the Fifth Edition of the Diagnostic and Statistical Manual of Mental Disorders (DSM-5) [6]. The partial remission qualifier was further specified by the DSM-IV definition of the residual phase whereby non-remitting features were restricted to “only negative symptoms or by two or more symptoms listed in Criterion A present in an attenuated form” [5]. Avolition was defined on the SCI-PANSS, requiring a rating of no less than 3 on the item labelled “disturbance of volition” (G13) as well as a minimum cumulative score of 10 for items G13, N4 (passive/apathetic social withdrawal) and N2 (emotional withdrawal) [18]. As the G13 item described as a “disturbance in the wilful initiation, sustenance, and control of one’s thoughts, behaviour, movements and speech” had previously been shown to overlap with expressive deficits (reflecting a loss of initiative) and social amotivation [19], this composite inclusive criterion was adopted for defining an avolitional population notwithstanding an overlap with other negative symptoms. Adequate information to this end was obtained through an interview and observation guide and scrutiny of clinical records, and information from nursing personnel and family. For inclusion, patients were furthermore required to be in a stable condition without any changes to their medication or dosages during the preceding three months, which was verified through self-report and the clinical records.

Exclusion criteria that pertained for the third data set were patients in an acute phase of schizophrenia as defined by DSM-5, and if self-reported or noted in the clinical records that there was a positive substance history during the preceding three months. Other exclusion criteria were intellectual disability, unstable or significant other medical disorders, and a past head injury with neurological sequelae or a loss of consciousness.

### 2.2. Assessments and Measuring Instruments

The 10-item Staden Schizophrenia Anxiety Rating Scale (S-SARS) being the focus of investigation in this study is a clinician-rated instrument for the assessment and measurement of specific and general anxiety in schizophrenia (see Appendix A). Five items of the specific anxiety subscale measure persecutory and nihilistic anxiety, perceptual anxiety, anxiety attacks, situational anxiety, and obsessive-compulsive anxiety. The five items of the general anxiety subscale measure somatic anxiety, psychomotor and cognitive agitation, worry and fear, control-related anxiety, and impairment from anxiety. Each item has six narrative anchor points to indicate severity of anxiety during the preceding week on a scale from 0 to 5, and is accompanied by guided questions for use during the interview as to inform the ratings.

Anxiety was also measured by the Hamilton Anxiety Rating Scale (HAM-A), which is one of the most established and widely used observer-rated scales for anxiety [20]. The HAM-A is a general measure of anxiety that is not specific to schizophrenia or any specified anxiety disorder. Its 14 items are each rated from zero (i.e., none) to four (i.e., severe, grossly disabling), which are mostly about somatic and behavioural manifestations of anxiety.

The Structured Clinical Interview for DSM-IV Axis I Disorders (SCID-I) was applied in clinician-administered interviews for diagnosing anxiety disorders [21,22].

The Structured Clinical Interview for the Positive and Negative Syndrome Scale (SCI-PANSS) served as measure of psychotic episode severity [18]. One of its items on anxiety, was analysed as an additional parameter of anxiety.

The Calgary Depression Scale for Schizophrenia (CDSS) comprises nine items with descriptive anchor points that are clinician-administered. Its validity is well-established in correlating strongly to very strongly with other measures of depression [23] and accurately distinguishing depressive features from negative symptoms and extrapyramidal side effects. Cronbach’s alpha coefficients between 0.7 and 0.9 suggest good internal consistency [24].

The Vigour Assessment Scale (VAS) measures by self-report both positive (being present) and negative (being absent) vigour [17]. Each of its 27 items are rated on a four-point Likert scale according to the frequency of the experience during the preceding seven days (1 = None of the time, 2 = Sometimes, 3 = Often, 4 = Most of the time). To obtain the total score, the subtotal of Category A (absence of vigour) is subtracted from the subtotal of Category B (presence of vigour). To prevent acquiescence bias, its positive and negative items are interspersed. Its convergent and discriminant validity were good, with a Cronbach’s alpha value of 0.82, a clear six-factor correlation structure and small error of measurement [17].

The Personal Growth and Initiative Scale (PGIS) measures the active involvement and development of an individual as a person. The 16-item version has four subscales, which are readiness for change (RC), planfulness (Plan), using resources (UR), and intentional behaviour (IB). It has a four-factor structure for which adequate goodness-of-fit indices were found. Adequate test-retest reliability indices ranged from 0.73 (UR) to 0.81 (Plan), and the internal consistency of the subscales was good (RC, α = 0.83; Plan, α = 0.84; UR, α = 0.80; IB, α = 0.89) [25].

The Behavioral Inhibition/Behavioral Activation Scales (BIS/BAS) measure an individual’s sensitivity to two motivational systems, captured by 24 items in four subscales [26]. Behaviour inhibition subscale (BIS) is about the anticipation of punishment, non-reward, and novelty. The three behaviour activation subscales are about drive (BAS-D), fun-seeking (BAS-FS), and reward responsiveness (BAS-RS). The subscales accounted cumulatively for 49% of the overall variance in a sample of 732 college students. Cronbach’s alpha values were as follows: BIS, α = 0.74; Reward Responsiveness, α = 0.73; Drive, α = 0.76; Fun Seeking, α = 0.66). Test-retest reliability after eight weeks in 113 subjects yielded moderately strong correlations. Schizophrenia patients in comparison with healthy controls showed no differences in BAS sensitivity but higher BIS sensitivity [27]. In a study among 151 schizophrenia patients regarding approach and avoidance tendencies, significant sensitivity patterns in behavioural inhibition and activation were found [28].

The Procrastination Scale (ProcS) comprises 20 items expressing a true-false measure on a Likert scale ranging from 1 to 5 [29]. Test-retest scores correlated strongly (r = 0.8) [30], and its Cronbach’s alpha was 0.71 [31].

The 10-item Fatigue Assessment Scale (FAS) uses a five-point Likert scale ranging from ‘never’ to ‘always’. It is not specifically designed for schizophrenia, but its validity and reliability have been shown for various other populations, including women with breast problems, construction workers, and mothers of infants and young children [32,33,34]. Internal consistency testing yielded alpha ratings between 0.88 and 0.90, and strong correlations in test-retest reliability measurements a month apart (r = 0.88). Discriminant validity testing was demonstrated for state anxiety, depressive features, and neuroticism [32].

### 2.3. Procedures and Ethics Approval

Patients and their clinical records were assessed for meeting the inclusion and exclusion criteria. The interview-based measures (these are the S-SARS, SCI-PANNS, SCID, HAM-A, and the CDSS) were administered by the authors, all suitably trained in their use. Each participant gave written informed consent to participate in the study, which was performed in accordance with the Declaration of Helsinki. Ethics approval was obtained from Faculty of Health Sciences Research Ethics Committee at the University of Pretoria, South Africa.

### 2.4. Statistical Analyses

Point-biserial correlation testing between S-SARS scores and gender was performed. Spearman’s rho and Kendall’s tau correlation coefficients were calculated for the correlation between the S-SARS and age, as the latter did not follow a normal distribution as is usually the case. Psychometric analyses were performed both for each data set and combined data sets insofar as measures were shared among the data sets, as the three data sets did not derive from an identical set of measures. In reporting the results, each table below indicates which measures were shared among the data sets and to which data sets the results pertain. For convergent validity, parametric (Pearson’s) correlations of the total S-SARS scores with the HAM-A and the anxiety item on the SCI-PANSS were tested and hypothesised to be moderate to strong. For criterion-related validity, a multiple canonical discriminant analysis was performed in examining the ability of an S-SARS model to account for three diagnostic groups as yielded by the SCID–these were, respectively, the no anxiety, the anxiety disorder not otherwise specified, and the specified anxiety groups. In addition, the ability of S-SARS model to accurately diagnose the presence or absence of an anxiety disorder was calculated in terms of specificity, sensitivity, positive predictive value, negative predictive value, miss rate or false negative rate, fall out or false positive rate, false discovery rate, and false omission rate using standard formulas for these. For discriminant validity, Pearson’s correlations between the S-SARS and each of the non-anxiety measures were tested and hypothesised to be no more than weak, or negative.

Reliability testing of the S-SARS comprises calculations of the Cronbach’s alpha coefficient for internal consistency, Spearman-Brown and Guttman split-half coefficients, and the Standard Error of Measurement (SEM) (as the product of the standard deviation and the square root of one minus the reliability coefficient). For structural validity and consistency of measurement, an exploratory factor analysis using Principal Axis Factoring was performed. The S-SARS’s inter-rater reliability was examined by calculating the extent of agreement among nine senior medical students who applied the S-SARS to an audio-visual recording of an interview with a patient who suffered from anxiety in the acute phase of schizophrenia. For the extent of agreement, the intra-class correlation coefficient and Krippendorff’s alpha for an ordinal metric were calculated.

The probability threshold for a type I error was set at 5%. The strength of correlation coefficients was defined as follows: r < 0.20 is negligible; 0.20 < r < 0.40 is weak; 0.40 < r < 0.60 is moderate; 0.60 < r < 0.80 is strong; r > 0.80 is very strong [35]. The strength of each psychometric property was defined conventionally as follows. Cronbach’s α > 0.9 is excellent, 0.8 < α < 0.9 is good, 0.7 < α < 0.8 is acceptable, 0.6 < α < 0.7 is questionable, 0.5 < α < 0.6 is poor, and α < 0.5 is unacceptable [8]. SPSS version 27 was used for the analyses.

## 3. Results

### 3.1. Descriptive Features

Table 1 presents the gender and age for three data sets comprising 343 participants, of whom three-quarters were male. The S-SARS did not correlate significantly with gender (r = 0.030; *p* = 0.580), but weakly and inversely with age (ρ = −0.209; *p* < 0.001; τ = −0.156; *p* < 0.001). Table 2 presents the mean scores, standard deviations, and 95% confidence intervals of the mean for the various measures in each of the data sets and when combined. Table 3 does the same, but for the measures that were only applied in the third data set. The mean S-SARS scores were higher for the in-patient data sets than for the outpatients (i.e., data set no 3), whereas the outpatients scored higher on the avolition items of the SCI-PANSS, which was consistent with the way in which the population for data set no 3 had been defined.

### 3.2. Convergent and Criterion-Related Validity

Supporting convergent validity, the S-SARS correlated strongly with the HAM-A (r = 0.711; see Table 4) and the anxiety item on the SCI-PANSS (r = 0.762). Furthermore, the S-SARS showed criterion-related validity with the diagnosis of an anxiety diagnosis yielded by the SCID. A multiple discriminant analysis on the first data set resulted in a two-function model for the S-SARS items that accounted for the three diagnostic groups, these are, respectively, the no anxiety, the anxiety disorder not otherwise specified, and the specified anxiety groups. With a high Eigenvalue of 3.7, Function 1 expressed 86.8% of the variance (*p* < 0.001; canonical correlation = 0.89), but Function 2 also contributed significantly (Eigenvalue = 0.56; 13.2% of the variance; canonical correlation = 0.6; *p* = 0.005). Figure 1 shows the diagnostic group centroids for the two functions, which depicts how well the two-function model of the S-SARS accounted for the three diagnostic groups. The structure matrix for the S-SARS items contributing to the standardised canonical functions is presented in Table 5.

As seen in Table 6, this S-SARS model correctly classified the patients into the three groups as classified by the SCID in 88.3% of cases. In total, 7 of the 60 patients were found to be misclassified; that is, their actual group differed from the group predicted by the S-SARS model. These all shared a very low rating on the 10th item of the S-SARS, which means that for all the misclassified cases, there was very little if any impairment owing to anxiety. Hence, these cases may be considered as “noise” to the model by less constitutive cases, presuming that little impairment implies less constitutive anxiety.

Instead of the three diagnostic groups, the ability of the S-SARS model to accurately diagnose the presence or absence of an anxiety disorder is reflected in the calculations presented in Table 7, based on the one false positive and two false negative instances of anxiety disorder yielded by the S-SARS model among 60 participants (as seen in Table 6).

### 3.3. Discriminant Validity

Supportive of the discriminant validity of the S-SARS, the correlation coefficients seen in Table 4 and Table 8 indicated at most a weak correlation (r < 0.5) of the S-SARS with each of the non-anxiety measures, except for the CDSS. The correlation coefficient for the CDSS (r = 0.722) when corrected for measurement error (using reliability coefficients of 0.93 and 0.8 for, respectively, the S-SARS and the CDSS) was reduced to 0.67, which suggests that discriminant validity likely exists between the two measuring instruments [36]. This was also supported by a statistical difference between the measures (t = 10.822, df = 352; *p* < 0.001).

### 3.4. Internal Consistency and Split-Half Reliability

As presented in Table 9, Cronbach’s alpha coefficients among the S-SARS items ranged from 0.875 to 0.931, indicating excellent internal consistency. Cronbach’s alpha coefficients for each half of the S-SARS were slightly lower, which is in keeping with the theoretical expectation that instruments comprising fewer items result in lower Cronbach’s alpha values. The Spearman-Brown and the Guttman Split-Half coefficients ranged from 0.864 to 0.937.

### 3.5. Standard Error of Measurement

As presented in Table 9, the standard error of measurement (SEM) for the total S-SARS score ranged from 1.54 to 2.58, subject to a 68% degree of certainty using one standard deviation, as is customary for SEM calculations. This means that within a theoretical range from 0 to 50, the observed total scores were within 1.54 to 3.58 points of the calculated true scores. In terms of internal consistency, this means that the total observed score consistently measured what it was supposed to measure, plus or minus 1.54 to 3.58 points.

### 3.6. Structural Validity and Consistency of Measurement

Preceding an exploratory factor analysis, the Kaiser–Meyer–Olkin test of sampling adequacy yielded a coefficient of 0.933, indicating the sample size was sufficient. A Bartlett’s test of sphericity was statistically significant (approximate chi-square = 2569.647, df = 45, *p* < 0.001), meaning that the S-SARS items were significantly related and suited for factor analysis.

The Kaiser criterion, by which factors with an eigenvalue greater than 1.0 should be retained, was applied when using principal axis factoring. This extracted a single factor after four iterations, which accounted for 62.25% of the variance (thus a satisfactory explanation) [37] with a factor score of 0.953 in the covariance matrix. The factor matrix for the S-SARS items is shown in Table 10, which suggests the unidimensionality of the S-SARS.

### 3.7. Inter-Rater Reliability

Nine raters were in agreement on the most common rating for each of the S-SARS items in 82.2% of ratings, which increased to 97.7% when agreement on the second most common rating was added. The intra-class correlation coefficient was 0.987, with a 95% confidence interval between 0.969 and 0.996. The Krippendorff’s alpha value was 0.852, which indicates strong agreement.

## 4. Discussion

In contrast with the findings of a recent review that no instrument measuring anxiety in non-affective psychoses exhibits strong psychometric properties and that all the evaluated instruments performed poorly against standardised quality assessment criteria [8], the results of the current study suggest that the S-SARS, as a clinician-administered measure of anxiety in schizophrenia, holds excellent psychometric properties. Criterion-related, convergent, discriminant, and structural validity testing showed that it measured what it was supposed to measure. The results also showed that it measured so consistently, as supported by good to excellent internal consistency and split-half reliability, substantial inter-rater reliability, and homogeneity of measurement.

Convergent validity of the S-SARS was found in its strong correlations with the anxiety item on the SCI-PANNS and another measure of anxiety (i.e., HAM-A), despite the HAM-A not being specific to the schizophrenia population and its narrower scope in focusing on somatic and behavioural manifestations of anxiety. Criterion-related validity modelling yielded accuracy, sensitivity, and specificity rates of around 95%. The discriminant validity of the S-SARS is supported by its correlating negatively and no more than weakly with behavioural drive, reward-responsiveness, fun-seeking behaviour, vigour, avolition, personal growth, and initiative. It correlated no more than weakly with behavioural inhibition, procrastination, and fatigue. The moderate correlations with the SCI-PANSS and akathisia and the strong correlation with depressive features are congruent with previous reports [9,11], but these were below the threshold required to indicate discriminant validity, supported by a statistically significant difference between the S-SARS scores and the measure of depressive features.

The S-SARS’s structural validity and consistency of measurement were confirmed in a single-factor unidimensional correlational model with a high factor score of 0.953, and 5 of the 10 items correlating very strongly, 3 items strongly, and 2 items moderately with the single factor. These results of the factor analysis suggest that the S-SARS measured a single construct and so measured what it was supposed to measure. These results also support the structural internal consistency of the measurement, which was congruent with the internal consistency reflected by good to excellent Cronbach’s alpha values and strong to very strong correlations between the halves of the S-SARS. The unidimensionality seen in the factor analysis reflects the internal structure of the S-SARS, whereas the two canonical discriminant functions reflect a two-dimensionality in the correlations of the S-SARS items with an external diagnostic criterion. Notwithstanding the S-SARS thus measuring a single unidimensional construct, the finding of its two-dimensional diagnostic property supports the premise that an anxiety measure of sufficient scope in the schizophrenia population should measure both specified and undifferentiated anxiety.

Homogeneity of the S-SARS’s measurements was found in it incurring in no more than a small error of measurement, whereby observed scores were within 1.54 to 3.58 of a true score on a scale ranging from 0 to 50. Consistency of measurements among raters was found in 82.2% of identical ratings for the most common rating for each item, a very high intra-class correlation coefficient of 0.987, and strong agreement among raters for all items, indicated by a Krippendorff’s alpha value of 0.852.

With these strong psychometric properties of the S-SARS, using it in research and clinical practice may improve the accuracy of measurement of anxiety in schizophrenia. The S-SARS measures anxiety as it features in both the various specified anxiety disorders and undifferentiated anxiety. It does not replace, however, measures with a narrower focus on one of the specified anxiety disorders. Measuring anxiety accurately in schizophrenia is important for various clinical and research reasons. Anxiety adds to the burden of schizophrenia by impacting negatively on quality of life [38], functioning [39,40], overall psychopathology, and the severity of comorbid medical conditions [41]. Increased rates of relapse, more frequent and longer duration of hospitalisations, poorer response to pharmacological treatments, substance abuse, negative attributional style, suicide, and suicide attempts have been associated with anxiety in schizophrenia [42,43,44,45].

Furthermore, claims have been made about the efficacy of antipsychotic medication in reducing anxiety in schizophrenia [46,47,48], but these are scientifically weak insofar as instruments measuring anxiety have been restricted in their scope and psychometric properties as described above. Using the S-SARS in efficacy studies may strengthen conclusions on the anxiolytic effects of anti-psychotic and other treatments in the schizophrenia population.

There are limitations pertaining to the results reported here, which may be addressed in further research. The inter-rater reliability parameters, although obtaining excellent values here, are rather tentative considering the small number of raters. This should be examined further among more raters that assess and rate several patients with varying degrees of anxiety. Furthermore, the outpatients in the study were limited to those with avolition. Although there were no more than weak correlations between anxiety and the avolitional parameter (see Table 4), anxiety qualities may be different in schizophrenia outpatients who are not avolitional. The male preponderance in this study is similar to that seen in probabilistic schizophrenia samples [49]. This study found no correlation between anxiety and gender, but the potential influence of females being under-represented needs to be studied further.

## 5. Conclusions

The strong psychometric properties reported here suggest that the S-SARS is a valid and reliable instrument for measuring specified and undifferentiated anxiety in the schizophrenia population. In contrast with existing anxiety measures not specific to schizophrenia, the S-SARS accounts for the anxiety that is expressed within delusional content and in disturbances of perceptions without conflating these symptoms. It provides the means in future studies for the accurate measurement of the anxiolytic effects of treatments, including antipsychotic medication, in the schizophrenia population.

## Figures and Tables

**Figure 1 diagnostics-12-00831-f001:**
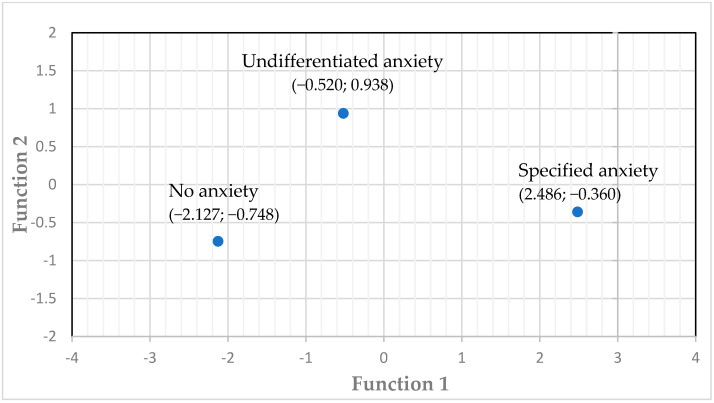
Diagnostic group centroids of the canonical discriminant functions.

**Table 1 diagnostics-12-00831-t001:** Descriptive characteristics of the participants.

Descriptive Feature		All Data Sets *n* = 353	Data Set No 1 *n* = 60	Data Set No 2 *n* = 51	Data Set No 3 *n* = 242
Gender	Male	74.5% *n* = 263	80% *n* = 48	76.5% *n* = 39	72.7% *n* = 176
	Female	25.5% *n* = 90	20% *n* = 12	23.5% *n* = 12	27.3% *n* = 66
Age	Minimum	18	19	19	18
	Maximum	64	60	64	62
	Mean	36.7	33.8	37.0	37.4
	Standard Deviation	11.0	10.4	13.3	10.6

**Table 2 diagnostics-12-00831-t002:** Descriptive statistics for the measures in combined data sets.

Instrument	Descriptive	Combined Data Sets	Data Set No 1 *n* = 60	Data Set No 2 *n* = 51	Data Set No 3 *n* = 242
Staden Schizophrenia Anxiety Rating Scale (S-SARS)	Mean	6.75	14.58	18.88	2.26
	SD	9.52	9.62	10.13	4.67
	95% CI ^1^ of the mean	5.79–7.75	12.28–16.82	16.12–21.64	1.66–2.85
Hamilton Anxiety Scale (HAM-A)	Mean	11.05	10.85	11.29	No data
	SD	9.03	10.68	6.68	No data
	95% CI of the mean	9.48–12.68	8.32–13.42	9.45–13.18	No data
Calgary Depressive Symptoms Scale (CDSS)	Mean	5.69	5.05	6.45	1.33
	SD	5.69	5.81	5.50	2.78
	95% CI of the mean	4.54–6.73	3.58–6.47	5.02–7.94	0.98–1.69
Barnes Akathisia Scale (BAS)	Mean	1.04	0.5	1.67	No data
	SD	1.69	1.2	1.96	No data
	95% CI of the mean	0.73–1.39	0.22–0.83	1.18–2.24	No data
Structured Clinical Interview for Positive and Negative Symptoms of Schizophrenia (SCI-PANSS)	Mean	100.76	103.43	97.61	No data
	SD	16.0	13.87	17.83	No data
	95% CI of the mean	97.78–103.92	99.95–106.77	92.83–102.59	No data
Avolitional items on the SCI-PANSS (G13, N2, N4 on SCI-PANSS)	Mean	12.65	10.28	10.18	13.76
	SD	3.40	2.85	2.99	3.01
	95% CI of the mean	12.29–13.0	9.52–11.02	9.33–10.94	13.38–14.14

^1^ CI = Confidence interval.

**Table 3 diagnostics-12-00831-t003:** Descriptive statistics for the instruments only in data set no. 3.

Instrument	Mean	Standard Deviation	95% Confidence Interval of the Mean
Vigour Assessment Scale (VAS)	12.02	25.42	8.81–15.24
Personal Growth and Initiative Scale-II (PGIS–II)	39.59	13.66	37.86–41.32
Behaviour inhibition (BIS)	19.82	4.38	19.19–20.37
Drive (BAS-D)	11.24	3.16	10.84–11.61
Reward-responsiveness (BAS-RS)	16.14	3.24	15.17–16.54
Fun-seeking (BAS-FS)	10.78	2.92	10.39–11.15
Procrastination Scale (ProcS)	56.64	12.17	55.1–58.2
Fatigue Assessment Scale (FAS)	24.96	8.70	23.86–26.06

**Table 4 diagnostics-12-00831-t004:** Pearson’s correlation coefficients among the measures in combined data sets.

Instruments	S-SARS *n* = 353	HAM-A *n* = 353	CDSS *n* = 353	BAS *n* = 111	SCI-PANSS *n* = 111	Avolitional Items on SCI-PANSS *n* = 111
S-SARS	1	0.711	0.722	0.418	0.455	−0.268
HAM-A	0.711	1	0.581	0.320	0.258	0.123
CDSS	0.722	0.581	1	0.368	0.407	0.315
BAS	0.418	0.320	0.368	1	0.235	0.190
SCI-PANSS	0.455	0.258	0.407	0.235	1	0.679
Avolitional items on SCI-PANSS	−0.268	0.123	0.315	0.190	0.679	1

**Table 5 diagnostics-12-00831-t005:** Structure matrix of the S-SARS items contributing to the standardised canonical functions.

^2^ S-SARS Item *n* = 60	^1^ Function 1	^1^ Function 2
Item 10: Impairment owing to anxiety	^3^ 0.652	0.271
Item 3: Anxiety attacks	^3^ 0.622	−0.461
Item 6: Somatic anxiety	^3^ 0.574	0.067
Item 8: Worry and fear	^3^ 0.457	0.331
Item 9: Control-related anxiety	^3^ 0.349	0.268
Item 4: Situational anxiety	^3^ 0.204	−0.168
Item 5: Obsessive-compulsive anxiety	^3^ 0.170	−0.081
Item 1: Persecutory and nihilistic anxiety	0.378	^3^ 0.532
Item 7: Psychomotor and cognitive agitation	0.221	^3^ 0.495
Item 2: Perceptual anxiety	0.295	^3^ 0.467

^1^ Pooled within-groups correlations between discriminating items and standardised, canonical discriminant functions. ^2^ Items ordered by absolute size of correlation within function. ^3^ Largest absolute correlation between each variable and any discriminant function.

**Table 6 diagnostics-12-00831-t006:** Diagnostic classification of the cases by the S-SARS discrimination model.

*n* = 60		SCID Diagnostic Groups	Predicted Group Membership			Total
			No anxiety	Anxiety disorder not otherwise specified	Specified anxiety disorder	
^1^ Original	Frequency	No anxiety	17	1	0	18
		Anxiety disorder not otherwise specified	2	19	1	22
		Specified anxiety disorder	0	3	17	20
	%	No anxiety	94.4	5.6	0	100
		Anxiety disorder not otherwise specified	9.1	86.4	4.5	100
		Specified anxiety disorder	0	15	85	100
^2,3^ Cross-Validated	Frequency	No anxiety	15	3	0	18
		Anxiety disorder not otherwise specified	5	14	3	22
		Specified anxiety disorder	0	4	16	20
	%	No anxiety	83.3	16.7	0	100
		Anxiety disorder not otherwise specified	22.7	63.6	13.6	100
		Specified anxiety disorder	0	20.0	80.0	100

^1^ 88.3% of original grouped cases correctly classified. ^2^ In cross validation, each case is classified by the functions derived from all cases other than that case. ^3^ 75.0% of cross-validated grouped cases correctly classified.

**Table 7 diagnostics-12-00831-t007:** Criterion-related validity calculations for the S-SARS model (*n* = 60).

Validity Calculation	Percentage
Accuracy	95%
Sensitivity	95.2%
Specificity	94.4%
Positive predictive value	97.6%
Negative predictive value	89.5%
Miss rate or false negative rate	4.8%
Fall out or false positive rate	5.6%
False discovery rate	2.4%
False omission rate	10.5%

**Table 8 diagnostics-12-00831-t008:** Pearson’s correlation coefficients among the measures only in data set no 3.

Measure	S-SARS	VAS	PGIS	BIS	BAS-D	BAS-RS	BAS-FS	ProcS	FAS
**S-SARS**	1	−0.279	−0.128	0.389	−0.127	−0.034	−0.147	0.268	0.335
**VAS**	−0.279	1	0.662	−0.045	0.531	0.542	0.421	−0.656	−0.684
**PGIS**	−0.128	0.662	1	0.127	0.577	0.583	0.513	−0.529	−0.492
**BIS**	0.389	−0.045	0.127	1	0.142	0.256	0.198	0.109	0.237
**BAS-D**	−0.127	0.531	0.577	0.142	1	0.676	0.642	−0.434	−0.376
**BAS-RS**	−0.034	0.542	0.583	0.256	0.676	1	0.678	−0.426	−0.389
**BAS-FS**	−0.147	0.421	0.513	0.198	0.642	0.678	1	−0.295	−0.288
**ProcS**	0.268	−0.656	−0.529	0.109	−0.434	−0.426	−0.295	1	0.618
**FAS**	0.335	−0.684	−0.492	0.237	−0.376	−0.389	−0.288	0.618	1

**Table 9 diagnostics-12-00831-t009:** Internal consistency of the S-SARS.

Internal Consistency Indicator		Combined Data Sets *n* = 353	Data Set No 1 *n* = 60	Data Set No 2 *n* = 51	Data Set No 3 *n* = 242
Cronbach’s alpha coefficient	Items 1–10	0.931	0.875	0.875	0.891
Split halves	Cronbach α (Items 1–5)	0.804	0.686	0.749	0.647
	Cronbach α (Items 6–10)	0.917	0.849	0.792	0.885
	Correlation coefficient	0.882	0.772	0.815	0.837
	Spearman–Brown coefficient	0.937	0.872	0.898	0.911
	Guttman Split-Half coefficient	0.928	0.864	0.897	0.874
Standard error of measurement (SEM)		2.50	3.40	3.58	1.54

**Table 10 diagnostics-12-00831-t010:** Factor matrix of the S-SARS items derived from a factor analysis (*n* = 353).

S-SARS Item	^1,2^ Single Factor
Item 10: Impairment owing to anxiety	0.893
Item 8: Worry and fear	0.891
Item 1: Persecutory and nihilistic anxiety	0.870
Item 9: Control-related anxiety	0.846
Item 6: Somatic anxiety	0.812
Item 2: Perceptual anxiety	0.781
Item 7: Psychomotor and cognitive agitation	0.732
Item 3: Anxiety attacks	0.731
Item 4: Situational anxiety	0.543
Item 5: Obsessive-compulsive anxiety	0.423

^1^ Extraction method: principal axis factoring. ^2^ One factor extracted after 4 iterations with a factor score of 0.953.

## Data Availability

The Research Ethics Committee that approved the research determines the limits on the availability of raw and processed data based on the merits of an application to gain access, the interests of stakeholders, and the mandates of research regulatory authorities. No special computer code or syntax is needed to reproduce analyses other than provided standardly in the SPSS-software program (version 27). Regarding the availability of research materials, the S-SARS is available as a supplement to the article.

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
