# Peer review of "Validity and Reliability of the Staden Schizophrenia Anxiety Rating Scale"

_diagnostics, 2022, doi:10.3390/diagnostics12040831_

Round 1
Reviewer 1 Report
The authors have provided a clear rationale and validation of their development of clinician ratings of 10 items about anxiety for use in people with schizophrenia whether they are acutely psychotic or in a residual state of avolition. The introduction, methods, analyses, and discussion are all excellent and detailed.
The main finding is that the 10 items have high internal consistency (0.9) as well as inter-rater reliability (.8). This means that the 10 items are all measuring nearly the same thing, which seems to be the overall level of functional impairment from anxiety across all the aspects of anxiety rated. Yet the discriminant analysis identified two significant functions in which the first explains 86% of variance and the second 14%, so there is some distinction between specific and non-specific anxiety.
It would be interesting to know the loadings on the two discriminant functions in order to understand more clearly whether the differentiation of specific and non-specific anxiety is strong enough to be useful, as suggested by the name of the scale. Then the authors could briefly discuss this issue. If I understand the intent of the scale, the goal is to provide clinicians with a rating tool to extract the overall level of functional impairment in schizophrenic patients from anxiety, whether that anxiety takes the form of a specific categorical diagnosis or is undifferentiated. So I am not sure whether the slight multidimensionality identified in the discriminant analysis should be viewed as a bonus that is too minor to be a weakness or a strength. Discussing this would clarify the goal sought and the practical outcome achieved.
Very minor typos: p4, 177 "self-report", not reprort
p4, 198, "in", not iIn
p14, 388 "so did", not did so
Table 2: data set 2 column heading needs reformatting
Table 10: note that missing item 5 is obsessive-compulsive anxiety?
Author Response
Dear Reviewer
Thank you for the careful reading and high quality engagement with our article. In response, we have added more details about the canonical discriminant analyses including the loadings and a figure that underscores your point well. We have also added the following text in the discussion: "The uni-dimensionality seen in the factor analysis reflects the internal structure of the S-SARS, whereas the two canonical discriminant functions reflect a two-dimensionality in the correlations of the S-SARS items with an external diagnostic criterion. Notwithstanding the S-SARS thus measuring a single uni-dimensional construct, the finding of its two-dimensional diagnostic property supports the premise that an anxiety measure of sufficient scope in the schizophrenia population should measure both specified and undifferentiated anxiety".
The typos and the line that got deleted from the one table have now been corrected.
The added value triggered by your comments is much appreciated.
Sincerely, the authors
Reviewer 2 Report
The article "Validity and reliability of the Staden Schizophrenia Anxiety Rating Scale" is devoted to a topical issue, is executed at a high level, and is of interest to the readers of the journal. Identified shortcomings:
- The main thing that I did not like was some confusion with three data sets. What is the difference between data sets 1 and 2? Why do these datasets need to be presented separately? Why is the assessment of discriminant validity carried out on the combined set (Table 4) and on set No. 3 (Table 8)? Why was the discriminant analysis carried out on 1 set? What dataset was used for factor analysis? Probably, splitting the data into 3 sets (plus a combined set) makes some sense, but this is not clear from the text of the article. In terms of study design, data sets 1 and 2 are more appropriately combined into one group "schizophrenia in the acute phase", which is significantly different from the group "residual avolitional schizophrenia". In this case, the analysis was performed on a larger group of patients, which would increase the statistical power. It may be more appropriate to perform statistical analysis on combined data, followed by additional analysis of differences in two clinically distinct subgroups ("schizophrenia in the acute phase" and "residual avolitional schizophrenia"). In general, authors need to clean up some confusion with datasets so that when reading the article it would be clear which datasets are being analyzed, in what cases and for what.
- You must indicate the years in which the data collection was carried out.
- Table 2 is mentioned in the text before table 1.
- 4. Table 5 can be removed by indicating all the data obtained in the text.
Author Response
Dear Reviewer
Thank you for the constructive comments. Responses are as follows:
1) Regarding the data sets, the article makes now clear that "Psychometric analyses were performed both for each data set and combined data sets insofar as measures were shared among the data sets, as the three data sets did not derive from an identical set of measures. In reporting the results, each table below indicates which measures were shared among the data sets and to which data sets the results pertain". The distinct data sets were thus not resulting from splitting the data, but each had originated from a distinct study (see end of introduction section). The difference between data set 1 and set 2 was that the SCID was performed for set 1 but not for set 2 or set 3. The canonical discriminant analyses towards criterion-related validity could thus only be performed on data set 1. Where two or three sets shared the same measure, the analyses were performed on the largest available body of data (i.e., thus combining data as much as possible) for that measure. This was done, for example, for the discriminant validity insofar all the measures were the same for all three data sets (reported in Table 4) whereas the measures in data set 3 (other than the S-SARS) do not feature in data sets 1 and 2, and for this reason Table 8 reports on the measures of data set 3. Where data were combined, however, the results are also reported for the individual data sets, as not to conceal/conflate potential differences between the data sets. The factor analysis was optimally done for all the data sets combined, and this is now clearly so indicated (as is the case for each table) - our apologies for not indicating this clearly in the initial manuscript.
2) The years during which data were collected, have now been inserted.
3) The mentioning of Table 2 before Table 1, has now been corrected by removing this first reference to Table 2 and refining the text by now referring to all the tables instead.
4) Table 5 has now been removed and its details are captured in the text.